# A Narrative of Oral Care in Palliative Patients

**DOI:** 10.3390/ijerph19106306

**Published:** 2022-05-23

**Authors:** Siri Flagestad Kvalheim, Gunhild Vesterhus Strand

**Affiliations:** Department of Clinical Dentistry, Faculty of Medicine, University of Bergen, Årstadveien 19, N-5009 Bergen, Norway; gunhild.strand@uib.no

**Keywords:** palliative care, oral dryness, xerostomia, oral diseases, lubrication, oral care procedures

## Abstract

Painful oral conditions represent a significant problem for most patients with a serious disease and palliative care needs. The main causes of such conditions may be associated with the underlying disease and its treatment, but primarily with adverse side effects of drugs. Oral conditions can lead to worsening of the illness and reduced quality of life. The objective of this narrative is to present an overview of oral care problems of this group, including present clinical practice. The evidence base for procedures and efficient products is weak. There is therefore an obvious need for research within this field.

## 1. Introduction

The field of palliative care is often referred to as the most interdisciplinary among medical disciplines. The field includes medical, mental, and spiritual issues. In addition to physicians and nurses, the treatment team in a palliative unit comprises psychologists, priests, physiotherapists, and occupational therapists, among others [1]. Dental care professionals, however, are most often absent. This is thought-provoking because mouth-related symptoms are among the most common ailments in the seriously ill and dying [2].

As a result of the rapid development of diagnostic aids and treatment methods throughout the last century, the average life expectancy has increased. Fewer people die from infections. More people grow old, often with complex diseases, including lifestyle diseases and dementia. Adverse conditions are common, most often after long-term use of several drugs and treatments such as chemotherapy and radiation. The change in life expectancy and complexity of diseases have made palliative care more demanding and advanced. Care previously undertaken in homes or private hospices has now to a greater extent been transferred to specialized nursing home units and hospital wards, which can provide suitable specialized medical care. However, medications, treatments, and chronic diseases themselves often lead to oral side effects such as dry mouth and candidiasis [3,4].

By the time the patient receives a life-threatening diagnosis, an impaired general condition has usually been manifested for some time. Fatigue and depression naturally influence personal oral hygiene measures; oral care and dental treatment may often be abandoned or neglected for weeks and months. One of the challenges for the implementation of oral health in palliative care is the organization of the health services; the patient’s private dentist is tied to their clinic and rarely accompanies the patient to the hospital. The hospital usually lacks a tradition for addressing oral issues with the consequence that oral symptoms are often neglected [5].

The evidence base for procedures and products used in oral care treatment of fragile patients in general and palliative in particular is weak. The objective of this narrative is to present the essential concepts associated with the most common oral conditions occurring in palliative patients as well as their consequences and present clinical practice.

## 2. Causes of Dry Mouth in Palliative Patients

A majority of patients at palliative care units are cancer patients [6]. Xerostomia, the subjective feeling of dry mouth, is common in patients who have received radiotherapy to the head and neck, with a prevalence of about 80% [7]. In other patients who receive chemotherapy, the prevalence of xerostomia has been estimated to be around 50% [3]. Medications used in palliative care, such as opioid analgesics, sedatives, antidepressants, anxiolytics, neuroleptics, and anticholinergics, all cause dry mouth [8]. In palliative patients, decreased fluid intake, diarrhea, fever, vomiting, and medication may affect the regulation of the salt–fluid balance and lead to dehydration. Mouth-breathing, which also causes xerostomia, is common in bedridden patients with little wake time [4].

## 3. Consequences of Oral Diseases in Palliative Patients

Many patients in palliative care have poor appetite. This tends to diminish further with decreasing salivary secretion [9]. A lack of saliva makes it difficult to moisten food, and dehydrated oral and pharyngeal mucous membranes make swallowing difficult. Furthermore, dysphagia is adversely affected by weakened mouth, tongue, and throat muscles, which is fairly common in this group of patients [10]. If oral care procedures are not performed, food residues can remain in the oral cavity for a long period of time.

Xerostomia makes speech more difficult because the tongue sticks to the mucosa, often leading to social withdrawal. Speech problems reduce patients’ ability to communicate with family and friends and may also prevent requesting help and relief from healthcare professionals [10].

Dry mouth, dehydration, and thirst are related symptoms. Registration of the occurrence of somatic symptoms in patients admitted to the palliative ward in Trondheim, Norway, showed that dry mouth was mentioned by 56% and thirst in 49% [2]. Frequent moisturizing may have a positive effect on the feeling of thirst [11].

Oral mucositis is a painful inflammatory condition that may occur in 80% of cancer patients who have undergone chemo- and radiotherapy. In the dehydrated oral mucosa of an immunosuppressed patient, the condition may cause ulceration. Mucositis can be so painful that it makes all oral intake impossible. Parenteral nutrition may then be considered. Treatment of both the above conditions involves some form of oral palliative care described below.

Another troublesome condition is oral candidiasis. In palliative care patients, the incidence has been estimated to 75% [12]. The condition can vary in appearance, in some cases as white scrapable coating, in other cases as redness. Predisposing factors for fungal infection are poor oral hygiene, hyposalivation, diabetes, certain drugs, a weakened immune system, and reduced nutritional status. The presence of dryness and fungi in the oral cavity and throat is often particularly troublesome. Infections, poor oral hygiene, and an empty stomach often lead to halitosis [4].

## 4. Oral Palliative Care

Although there is no international consensus on procedures in oral palliative care, assessment and clinical procedures for oral care occur, to a large extent, fortuitously. There are no generally accepted and validated national guidelines. If they do exist, they are often questionable due to a lack of scientific evidence. There are large divergences between various procedures regarding frequency and choice of oral care products. In Norway, a group linked to the Regional Centre of Excellence for Palliative Care has developed a method for assessing the oral status and proposed procedures for oral care to palliative patients (Table 1). This contains a systematic checklist of items to be considered: which instruments/products to use for the various procedures and also a specification of the different oral care procedures that must be implemented. However, as stated above, these and other similar guidelines have largely emerged through tradition and extensive practical experience and based not on scientific evidence. The basic premise is that sanitation and hygiene are important starting points for good health. (Figure 1).

## 5. Assessment of Oral Disease

The accurate assessment of clinical signs and symptoms is a prerequisite for good treatment. The Edmonton Symptom Assessment System (ESAS) is the most widely used instrument for assessing symptoms in palliative medicine. When the form was first translated and used in Norway at the Palliative Medicine Unit, Trondheim Regional Hospital in 1999, the item “oral dryness” was added to the form. This version, known as the Trondheim Palliative Assessment Tool (T-PAT), became widely used in Norway, and was, until 2012, recommended and included in the national palliative care program.

In the revised form (ESAS-r), questions on oral dryness have been removed [13]; it is therefore unsuitable for the assessment of oral symptoms. At any rate, despite extensive use of the ESAS, studies have shown that patients may misunderstand both terminology and numerical values. Furthermore, ESAS is unsuitable for dementia patients [14], who are often unable to express pain due to reduced memory and language proficiency [15].

MOBID 2 (Mobilisation-Observation-Behaviour-Intensity-Dementia) is then more appropriate, as it is specifically designed for patients with dementia. Mobid-2 can detect pain based on observations in relation to behavioral change, which is recorded when the patient is mobilized. This is the only tool that covers the whole body when recording location and intensity. Pain sounds, facial expressions, and averting reactions are registered by ticking. Objectively observed pain behavior is additionally interpreted using a 10-point VAS scale [16]. The Revised Oral Assessment Guide (ROAG) is specifically developed for oral assessment. It has high sensitivity and specificity to assess the voice and condition of the lips, oral mucosa, tongue, gums, teeth, prostheses, and saliva. Oral symptoms are graded 1–3, where measures are to be taken at grade 2 or 3 [17].

## 6. Lip Lubrication

The oral care procedure is initiated by lubricating the lips. The purpose is to avoid cracking and painful wounds. In order to avoid infection, it is recommended to use lip cream rather than lip balm from a tube. Whether the use of cream rather than a tube has any practical significance for patients who do not have infected wounds in the oral cavity is uncertain. It has been suggested that petroleum lip balms should be discouraged due to possible flammability and aspiration risks [18].

## 7. Cleaning of Teeth and Mucosa

The purpose of mucosal cleansing in palliative care is to remove coatings, prevent infections and thereby reduce pain. It is recommended that the teeth are brushed with a soft brush and with toothpaste without sodium lauryl sulphate (SLS) [19,20]. Oral care should be specifically tailored to the intended purpose. Rather than slavishly following a set of fixed procedures, the patient’s symptoms and actual needs must govern the treatment.

In a healthy person, cleansing of the oral cavity is achieved by movement of the tongue cheeks, lips, and masticatory muscles, the cleansing effect of saliva and other liquids, and daily brushing. Daily use of bactericidal agents should be avoided in healthy individuals in order to maintain normal oral flora and because bactericidal agents may damage the oral mucosa [21]. In patients with oral disorders, such products may nevertheless be recommended for removing crusts, fungi, bacteria, food debris, and thick saliva. For that purpose, a procedure recommended by Western Norway Regional Health Authority [13] suggests cleansing with 0.5% hydrogen peroxide or NaCl 9 mg/ml [17,18,19]. However, it is not known what the long-term effect of active solutions such as hydrogen peroxide, applied frequently, may have on the oral mucosa. Saline (NaCl 9 ml/mL) has the same osmotic concentration as the tissue fluids and is therefore used as an infusion fluid and wound cleanser. It is also frequently used to clean the mucosa. However, cleaning with pure water or a more dilute saline solution might be equally beneficial and sufficient in cases when there are no clear signs of infection or a pronounced presence of crusts and coatings.

## 8. Mucosal Lubrication

A study of a representative sample of Norwegian hospitals and nursing homes showed that there is great variation in the procedures for oral care. Thus, regarding lubrication, more than 20 different products were identified: cream, cooking oils, special mixtures (e.g., Düssedorf mixture) and commercial products sold by pharmacies. No strong evidence exists indicating that any of these topical agents are effective per se in relieving the symptom of dry mouth. However, it has been documented that oral care with lubrication provides relief if care is provided frequently [22,23].

There are several saliva substitutes, rinses, gels, and sprays that are claimed to counteract dry mouth. They often mimic the appearance and flow properties of saliva but do not have similar enzymatic, antimicrobial, antifungal, or antiviral properties, nor do they have the physical properties of saliva. Thus, for example, the surface tension of the substitutes may cause the remaining saliva to be displaced [24]. Choosing the right lubricating product can therefore be a challenge. It may be expedient to let the patient try different products. For patients in palliative care who tend to have nausea, both the consistency and taste of the oral care products can cause problems. Products with a mild taste and which are not too viscous may then be more appropriate [22].

The use of glycerol in various forms is widespread. Glycerol solutions have been used in oral palliative care in many countries since the 1950s. However, in some countries, the use of glycerol is discouraged because it may have a desiccating rather than a moisturizing effect [18,25]. By contrast, a 17% concentration of glycerol has been recommended [19]. This solution provides good, immediate relief, but the effect disappears after a short period of time [22]. The long-term effect of this glycerol concentration on mucous membranes is unknown, but it seems unlikely that short-term use would cause problems. If used undiluted, glycerol may cause damage and discomfort and result in substantial worsening of dry mouth. Unfortunately, pure (85%) glycerol is occasionally used, probably due to unclear procedures or a lack of knowledge [5]. If a glycerol solution is to be used, the concentrated glycerol must be mixed with water in a 1:4 ratio and applied frequently [19,22].

Some patients with dry mouth prefer products that stimulate secretion, such as lozenges and chewing gum. If these are to have an effect, the salivary glands must have some residual secretion. Palliative patients with severe oral dryness may have difficulties dissolving lozenges. Chewing gum, intended for the same purpose, may also be unsuitable because palliative patients often lack chewing force [4].

Pilocarpine is a muscarinic receptor agonist that increases secretion from exocrine glands and can provide effective treatment for xerostomia with a dose of at least 20 mg per day [26]. However, its use involves a number of undesirable side effects such as sweating, headaches, urination, and vasodilation. There is currently little experience with the use of Pilocarpine in oral palliative care. The side effects represent an obvious challenge, especially if Pilocarpine is combined with other drugs.

Artificial hydration in palliative patients may reduce symptoms of dry mouth, but such a treatment is controversial because it is likely to affect fluid retention and cause pulmonary oedema. In addition, moral, ethical, and cultural problems may arise with this treatment [27].

For terminal patients, oral care as previously outlined, including symptom assessment, cleaning, and lubrication, will normally be less relevant. During the very last days of life, the main emphasis must be on regular moistening of the mucous membranes.

## 9. Discussion

There is general agreement on the importance of having a systematic and evidence-based approach to oral palliative care. Nevertheless, this field has received little scientific attention. To our knowledge, no systematic reviews or meta-analyses exist. For example, there is scant scientific evidence indicating that one oral care product is better than another [28]. Most clinical treatments are therefore experience-based.

Oral palliative care is in the border area between dentistry and medicine, and the two disciplines therefore have a shared responsibility for the treatment. The need for interdisciplinary collaboration is particularly evident when treatment and medications with adverse effects on oral conditions are dispensed. Awareness of such problems should be recognized by both professions and steps taken to alleviate them.

Collaboration is also important for research aimed at developing and improving treatments. Oral care must be recognized as a necessary daily care, in line with any other type of medical care. It must be assured that a lack of awareness of oral palliative care in healthcare professionals does not prevent patients from accessing symptomatic treatment based on measures that are the best known to date.

## 10. Conclusions

There are many challenges in providing care for dying persons. A warm heart and the best of intentions are not always enough. There are many methodological problems associated with research within this area that must be overcome. Medical, dental, and other health professionals should form interdisciplinary consensus groups in order to develop evidence-based, practical guidelines for optimized palliative care.

## Figures and Tables

**Figure 1 ijerph-19-06306-f001:**
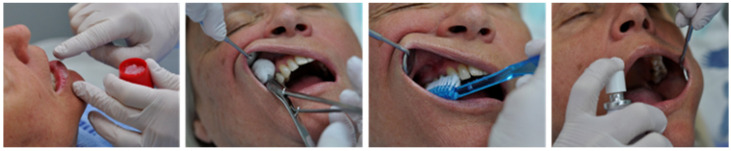
Illustration of oral care procedure: (1) lubrication of lips, (2) cleaning of mucous membranes, (3) cleaning of teeth and (4) lubrication of mucous membranes, here using a moisturizing spray.

**Table 1 ijerph-19-06306-t001:** Procedures for oral care in palliative patients, prepared by a group from the Regional Centre of Excellence for Palliative Care, Norway (translated into English by the authors of this article).

Oral Care for Palliative Patients—The Short Version		
If the Patient Is Unconscious, Dentures Must Always Be Removed. Special Care Must Be Taken When Oral Care Is Carried Out		
Item to Be Considered	Instruments/Products	Oral Care Procedures
Hygiene, general measures	Gloves—non-sterile	Hand cleaning is performed before and after the oral care procedure. Oral care is a clean procedure, but sterile gloves are usually not required. Procedures must be individualized.
Preparation, before procedures	Mouth mirror or spatula, light or torch (flashlight), gloves, disposable kidney bowl, towel/paper	Inform patient. Position patient as well as possible to complete the oral care procedure.
Oral assessment	Oral assessment form ROAG (The Revised Oral Assessment Guide)	Examine lips, tongue, teeth, gums, and mucous membrane. Examine signs and symptoms from the oral cavity, routinely and systematically. Should be carried out at admission, thereafter when needed. Record findings in journal.
Lubrication of lips	White Vaseline or water-based lip cream	Always start and end with lip lubrication.
Cleaning of teeth and mucous membranes	Soft toothbrush with small head, preferably electric toothbrush. Toothpaste with neutral soap, small amount. Lockable tweezers with gauze. Hydrogen peroxide 0.5% (hydrogen peroxide 3% 15 mL in 75 mL water); alternatively, NaCl 9 mg/mL.	Clean teeth, tongue, and oral mucosa. Dentures must be removed and cleaned. After brushing teeth, tongue and mucous membranes are cleaned with 0.5% hydrogen peroxide. Should be performed twice daily, more often if necessary.
Lubrication of mucosa	Glycerol 17% (glycerol 85%: 10 mL in 40 mL water) or moisturizers according to the patient’s preference	Apply lubricating agent on mucosa, gingiva, and tongue.
Hygiene measures, after oral care procedures	Clean water instrument washer	Toothbrushes are rinsed under running water after use, air dried standing and stored dry and clean. Remove disposable equipment immediately after use. Reusable equipment is cleaned in tap water, disinfected in washing machine between each patient. Store dry.
Documentation	Patient journal assessment form	Relevant information concerning the patient’s oral health must be registered in the patient’s medical record to provide proper health care.

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
