# Peer review of "A Narrative of Oral Care in Palliative Patients"

_ijerph, 2022, doi:10.3390/ijerph19106306_

Round 1
Reviewer 1 Report
This manuscript is an original article which introduced oral symptoms and care in palliative patients. It includes informative information for clinicians in this field.
However, I have serious concerns with this manuscript.
Major
- The construction is somewhat disorderly and confusing. I recommend the authors describe characteristics (prevalence, causes, consequences, and treatment) of mucositis in addition to those of dry mouth, as those two symptoms seems different conditions.
Minor
- This manuscript seems a mini-review rather than an original article.
- Table 1 is hard to read. Please modify it and add subtitles on each column.
- I recommend that authors explain oral palliative care along the descriptions in Table 1.
Author Response
We are very thankful for reviewer’s constructive criticism.
- The construction is somewhat disorderly and confusing. I recommend the authors describe characteristics (prevalence, causes, consequences, and treatment) of mucositis in addition to those of dry mouth, as those two symptoms seems different conditions.
Response: We have added the prevalence and main cause of mucositis. Description of the consequences of this condition occurs in the original text.
- This manuscript seems a mini-review rather than an original article.
Response: We agree. Due to the scant evidence base of oral care in palliative patients, this paper should be identified as a clinical narrative. The title and relevant passages are therefore altered accordingly.
- Table 1 is hard to read. Please modify it and add subtitles on each column.
Response: We agree. Table 1 has been modified as suggested. (Updated table is sent as a separate attachment in addition to the manuscript.)
- I recommend that authors explain oral palliative care along the descriptions in Table 1.
Response: Oral palliative care procedures are now identified in extreme right column.
Reviewer 2 Report
Review
I was very pleased to review this manuscript on a very important topic. As stated by the authors, the evidence base for different procedures and agents is very scarce and it is emphasized throughout the manuscript.
I have two remarks
Please state in the title that this is a narrative review
References for MOBID and ROAD tools need to be listed. Add some more details about these two scales. What is assessed in each scale and how are they scored?
Author Response
Reviewer 2:
We are very thankful for reviewer’s constructive criticism.
I was very pleased to review this manuscript on a very important topic. As stated by the authors, the evidence base for different procedures and agents is very scarce and it is emphasized throughout the manuscript. I have two remarks
- Please state in the title that this is a narrative review
Response: This is now indicated – in the title and text
- References for MOBID and ROAD tools need to be listed. Add some more details about these two scales. What is assessed in each scale and how are they scored?
Response: Thank you for pointing out that reference is missing. References have now been inserted and details of the two scales have been added.
Reviewer 3 Report
It is important to add
1.- Periodontitis (by lack of oral hygiene) as risk factor to cardiovascular and pulmonary diseases, example Infective Endocarditis and Pneumonia by Aspiration.
2.- Vitamin B12 deficiency, presents smooth tongue.
3.- Vitamin B6 deficiency, presents inflammation of the tongue, sores or ulcers of the mouth and ulcers of the skin at the corners of the mouth.
4.- If the patient presents xerostomy, the team of M.D must look for xerophthalmia, because, could be present an immunologic disease.
5.- I suggest to write the importance of cleaning prothesis too.
Bibliography:-
The number 21. Doesn´t have year of publication.
Author Response
Reviewer 3:
We are very thankful for reviewer’s constructive criticism.
- Introduction can be improved
Response: This statement is unqualified, and it is therefore difficult to know specifically what the reviewer means. No change in the text has been made.
It is important to add
- Periodontitis (by lack of oral hygiene) as risk factor to cardiovascular and pulmonary diseases, example Infective Endocarditis and Pneumonia by Aspiration.
- - Vitamin B12 deficiency, presents smooth tongue.
- Vitamin B6 deficiency, presents inflammation of the tongue, sores or ulcers of the mouth and ulcers of the skin at the corners of the mouth.
- If the patient presents xerostomy, the team of M.D must look for xerophthalmia, because, could be present an immunologic disease.
Response: We respectfully disagree with the above points. The subject discussed in this article is limited to aspects of oral care in palliative patients. In our brief clinical overview, we have mentioned the main causes and consequences of diseases that occur frequently in this very special group of patients. However, we do not claim to have mentioned all possible factors and conditions, like the more seldomly occurring ones mentioned by the reviewer, that might be associated with oral diseases in general. If we were to do so, that would constitute a completely different article. No change in the text has been made.
- I suggest to write the importance of cleaning prothesis too.
Response: The importance of cleaning dental prostheses is specified in Table 1. No change in the text has been made.
- Bibliography: The number 21. Doesn´t have year of publication.
Response: We are grateful to the reviewer for spotting this error. The correct year (2021) has been added.
Reviewer 4 Report
The paper discusses oral care for patients in palliative care. There are some issues. The paper needs to be revised.
First, this is NOT an article but seems to be a narrative review. The authors should revise the whole text following the guideline and add a Narrative Review Checklist. Clinical question is unclear. Furthermore, the important categories in the check list are lack.
Second, references are insufficient. There are many reports in this field. Please refer recent papers.
Third, the area is not within the scope of this journal. Please transfer the paper to dental journal.
Author Response
Reviewer 4:
We are very thankful for reviewer’s constructive criticism.
The paper discusses oral care for patients in palliative care. There are some issues. The paper needs to be revised.
Response: The paper is now revised.
- First, this is NOT an article but seems to be a narrative review. The authors should revise the whole text following the guideline and add a Narrative Review Checklist. Clinical question is unclear. Furthermore, the important categories in the check list are lack.
Response: Our study does not pretend to be an experimental study. In accordance with the suggestion made by the reviewer, we have now unequivocally defined our paper as a narrative in title as well as in the text. We have consulted the Narrative Review Checklist, and revised the text accordingly.
The claim that the clinical question is unclear is unspecified, and it is therefore difficult to know specifically what the reviewer means. No change in the text has been made.
- Second, references are insufficient. There are many reports in this field. Please refer recent papers.
Response: We respectfully disagree. Our extensive search of relevant literature has not yielded much. This is documented in our recent publication (2019 reference number 20). Also, the claim is unspecified, and it is therefore difficult to know specifically what the reviewer means. No change in the text has been made.
- Third, the area is not within the scope of this journal. Please transfer the paper to dental journal.
Response: We respectfully disagree. One of the points defining the aims and scope of this journal is: “Health Care Sciences and Services”, which precisely what our paper is about. No change in the text has been made.
Reviewer 5 Report
Dear Authors,
Thank you for the effort made to produce this article. However, I would like to send you some considerations.
- The abstract shows a social rather than a scientific situation of the situation of patients receiving palliative treatment and its relation to oral health. However, there is no scientific data and there is no sample size. It also does not say whether the structure is a literature review or the type of study.
It does not discuss the methodology used to produce the article, but simply discusses side effects of treatment in patients with severe diseases and their consequences on oral health. Nor are these treatments described by referencing systematic reviews or meta-analyses.
In the discussion, it does not provide data or compare articles that speak to the above, so it is insufficient.
In my opinion, they should propose a scientific methodology of the subject to be dealt with and follow a suitable structure to be able to present it in scientific journals, as what they have sent is similar to a book, not a scientific article.
Best Regards
Author Response
Reviewer 5:
We are very thankful for reviewer’s constructive criticism.
Dear Authors,
Thank you for the effort made to produce this article. However, I would like to send you some considerations.
- Conclusions can be improved.
Response: This comment is unspecified, and it is therefore difficult to know specifically what the reviewer means. However, there are some changes made in both abstract and discussion and summary in the revised manuscript (marked in red) that might constitute the desired improvement.
- The abstract shows a social rather than a scientific situation of the situation of patients receiving palliative treatment and its relation to oral health. However, there is no scientific data and there is no sample size. It also does not say whether the structure is a literature review or the type of study.
Response: We do not see that the abstract contains any mention of social aspects. By contrast, such matters are briefly mentioned in the introduction and elsewhere in the text – simply because, in our clinical experience, they play a major role in the entire disease picture. No change in the text has been made.
- It does not discuss the methodology used to produce the article, but simply discusses side effects of treatment in patients with severe diseases and their consequences on oral health. Nor are these treatments described by referencing systematic reviews or meta-analyses.
Response: Our paper does not pretend to be a conventional scientific paper, with associated stringent methodology. Its purpose is to provide a narrative of a clinical problem that has received little scientific attention in the past, what aspects are important, and how the problem is managed presently. To underline this fact, we have changed the title and text. The reason why we have not referred to systematic reviews or meta-analyses is that, to our knowledge, these do not exist in dental literature.
- In the discussion, it does not provide data or compare articles that speak to the above, so it is insufficient.
Response: For the reason stated above, data and comparisons cannot be made in the discussion. No change in the text has been made.
- In my opinion, they should propose a scientific methodology of the subject to be dealt with and follow a suitable structure to be able to present it in scientific journals, as what they have sent is similar to a book, not a scientific article.
Response: We respectfully disagree. Papers that present clinical problems and how they are managed in clinical practice are certainly within the scope of a scientific journal. In fact, in our opinion, such articles are among the most read.
Reviewer 6 Report
please rewrite this article as either a scoping review, if the evidence in the form RCT is very limited, or a systematic review
abstract very should and has no structure, objectives, methods/material, conclusion,
no objective was defined for the study, add this at the end of introduction,
device your review search strategy,
a very short discussion and introduction, no conclusion!!
Author Response
Reviewer 6:
We are very thankful for reviewer’s constructive criticism.
- please rewrite this article as either a scoping review, if the evidence in the form RCT is very limited, or a systematic review
Response: We agree. Due to the scant evidence base of oral care in palliative patients, this paper should be identified as a clinical narrative. The title and relevant passages are therefore altered accordingly.
- abstract very should and has no structure, objectives, methods/material, conclusion,
Response: We have now unequivocally defined our paper as a clinical narrative in title as well as in the text. We have consulted the Narrative Review Checklist, and revised the text of the abstract and elsewhere accordingly.
- no objective was defined for the study, add this at the end of introduction,
Response: This has now been added to the end of the introduction.
- device your review search strategy
Response: We respectfully disagree. As our paper is in fact a clinical narrative, no research strategy is required or would be meaningful. No change in the text has been made.
- a very short discussion and introduction, no conclusion!!
Response: In a clinical narrative, the introduction and discussion and conclusion should contain all essential elements. If so, concise writing is an advantage rather than a disadvantage. We disagree that the discussion has no conclusion. In fact, the last sentence it is stated that: “Medical, dental and other health professionals should form interdisciplinary consensus groups in order to develop evidence based, practical guidelines for optimized palliative care.”
Round 2
Reviewer 1 Report
I appreciate revising the manuscript according to my suggestion. The revised manuscript is much improved. However, the following minor issue require clarification:
- I recommend that authors explain oral palliative care in the text based on the descriptions in Table 1.
Author Response
Reviewer 1
We would like to thank the reviewer once again for a conscientious review and for the opinions and suggestions made.
- I appreciate revising the manuscript according to my suggestion. The revised manuscript is much improved. However, the following minor issue require clarification:
Response: We are pleased that the reviewer finds our revised article much improved.
- I recommend that authors explain oral palliative care in the text based on the descriptions in Table 1.
Response: We have been somewhat uncertain as to what the reviewer means. In the section entitled “Oral palliative car” there is an extensive description of this concept. However, we have come to the conclusion that the reviewer probably suggests we add to the text the gist of our systematic approach to the concept embodied in Table 1. This we have now done (lines 106 – 109).
Reviewer 5 Report
The article has changed its structure from a systematic review to a narrative. In my opinion, this is the right way to publish this type of article, but the contribution and innovation of the article is still low. Congratulations for the effort.
Best Regards
Author Response
Reviewer 5
We would like to thank the reviewer once again for a conscientious review and for his/ her opinions.
- The article has changed its structure from a systematic review to a narrative. In my opinion, this is the right way to publish this type of article.
Response: As stated in our previous response, we agree that this type of article should be presented as a clinical narrative, and are pleased the reviewer appears to be satisfied with the changes made.
- but the contribution and innovation of the article is still low.
Response: We respectfully disagree with the claim that the contribution and innovation of the article is low. In our experience, down-to-earth articles that describe status quo within a clinical field, and particularly within a field that is almost devoid of scientific evidence, is of considerable value for clinicians and researchers alike.
- Congratulations for the effort.
We graciously accept the congratulation for our effort.
Reviewer 6 Report
thank you for your revision, however, i don't think paper offers anything new , as i previously mentioned this should be in a form of systematic review
Author Response
Reviewer 6
We would like to thank the reviewer once again for a conscientious review and for his/ her opinions.
- I thank you for your revision, however, I don't think paper offers anything new
Response: We respectfully disagree that our paper does not offer anything new. True, it does not contain any new scientific evidence on which to base clinical practice. However, in our experience, down-to-earth articles that describe status quo within a clinical field, and particularly within a field that is almost devoid of scientific evidence, is of considerable value for clinicians and researchers alike.
- as I previously mentioned this should be in a form of systematic review
Response: We respectfully disagree that our paper should be in a form of systematic review. As argued previously, one absolute condition for doing so, is that there is a considerable body of scientific evidence within the specific field on which to conduct such a review. To our knowledge, no such evidence base exists with regard to oral care in palliative patients.